# RNA-binding protein Elavl1/HuR is required for maintenance of cranial neural crest specification

Erica J Hutchins[1,2]*, Shashank Gandhi[3], Jose Chacon[4], Michael Piacentino[1], Marianne E Bronner[1]

[1]Division of Biology and Biological Engineering, California Institute of Technology, Pasadena, United States; [2]Department of Cell and Tissue Biology, University of California, San Francisco, San Francisco, United States; [3]The Miller Institute for Basic Research in Science, University of California, Berkeley, Berkeley, United States; [4]Department of Biology, School of Math and Science, California State University Northridge, Northridge, United States

**Abstract** While neural crest development is known to be transcriptionally controlled via sequential activation of gene regulatory networks (GRNs), recent evidence increasingly implicates a role for post-transcriptional regulation in modulating the output of these regulatory circuits. Using available single-cell RNA-sequencing datasets from avian embryos to identify potential post-transcriptional regulators, we found that *Elavl1*, which encodes for an RNA-binding protein with roles in transcript stability, was enriched in the premigratory cranial neural crest. Perturbation of Elavl1 resulted in premature neural crest delamination from the neural tube as well as significant reduction in transcripts associated with the neural crest specification GRN, phenotypes that are also observed with downregulation of the canonical Wnt inhibitor *Draxin*. That *Draxin* is the primary target for stabilization by Elavl1 during cranial neural crest specification was shown by RNA-sequencing, RNA immunoprecipitation, RNA decay measurement, and proximity ligation assays, further supporting the idea that the downregulation of neural crest specifier expression upon Elavl1 knockdown was largely due to loss of *Draxin*. Importantly, exogenous *Draxin* rescued cranial neural crest specification defects observed with Elavl1 knockdown. Thus, Elavl1 plays a critical a role in the maintenance of cranial neural crest specification via *Draxin* mRNA stabilization. Together, these data highlight an important intersection of post-transcriptional regulation with modulation of the neural crest specification GRN.

*For correspondence:
erica.hutchins@ucsf.edu

## Editor's evaluation

In this short report, Hutchins et al. reveal expression of the RNA-binding protein (RBP) HuR in the neural tube and cranial neural crest of chicken embryos. Knock-down of HuR affects expression of Axud1 and FoxD3 (both genes associated with neural crest specification) and of the Wnt antagonist Draxin previously shown by the authors to regulate neural crest specification and delamination. The authors propose that HuR associates with Draxin mRNA and demonstrate that Draxin overexpression can rescue FoxD3 expression upon HuR knock down. The data is in line with the idea that control of neural crest specification by HuR at least partially involves Draxin mRNA stabilization.

## Introduction

Neural crest cells are an essential, multipotent cell population in the vertebrate embryo. During development, these cells undergo coordinated induction, specification, and epithelial–mesenchymal

**eLife digest** As an embryo develops, different genetic programs become activated to give cell populations a specific biological identity that will shape their fate. For instance, when certain sets of genes get switched on, cells from the outermost layer of the embryo start to migrate to their final destination within the body. There, these 'neural crest cells' will contribute to bones and cartilage in the face, pigmented skin spots, and muscles or nerves in the gut.

When genes responsible for the neural crest identity are active, their instructions are copied into an 'RNA molecule' which will then relay this information to protein-building structures. How well the RNA can pass on the message depends on how long it persists within the cell. Certain RNA-binding proteins can control this process, but it is unclear whether and how this regulation takes place in neural crest cells. In their work, Hutchins et al. therefore focused on identifying RNA-binding proteins involved in neural crest identity.

Exploratory searches of genetic data from chick embryos revealed that, even before they started to migrate, neural crest cells which have recently acquired their identity produced large amounts of the RNA-binding protein Elavl1. In addition, these cells did not behave normally when embryos were deprived of the protein: they left the outer layer too soon and then switched off genes important for their identity. Genetic studies of neural crest cells lacking Elavl1 revealed that this effect was due to having lost the RNA molecule produced from the Draxin gene.

Introducing an additional source of Draxin into mutant embryos missing Elavl1 was enough to restore normal neural crest behaviour. Further biochemical experiments then showed that the RNA for Draxin decayed quickly in the absence of Elavl1. This suggests that the protein normally allows Draxin's RNA to persist long enough to pass on its message.

These results reveal a new mechanism controlling the identity and behaviour of the neural crest. Since many cancers in adulthood arise from the descendants of neural crest cells, Hutchins et al. hope that this knowledge could lead to improved therapies in the future.

transition (EMT) events to migrate and ultimately form a myriad of tissues, including craniofacial structures, components of the peripheral nervous system, as well as many other derivatives (*Gandhi and Bronner, 2018*). The transcriptional control of these events has been dissected and mapped into modules of a feed-forward gene regulatory network (GRN), which helps explain the detailed sequence of events involved in neural crest development (*Martik and Bronner, 2017*; *Simões-Costa and Bronner, 2015a*; *Williams et al., 2019*). Recently, in addition to transcriptional events, there has been growing appreciation for the role that post-transcriptional regulation plays in the establishment, maintenance, and regulation of neural crest formation (*Bhattacharya et al., 2018*; *Cibi et al., 2019*; *Copeland and Simoes-Costa, 2020*; *Forman et al., 2021*; *Sánchez-Vásquez et al., 2019*; *Ward et al., 2018*; *Weiner, 2018*).

Given that RNA-binding proteins play an essential role in post-transcriptional regulatory processes (*Dassi, 2017*), we sought to broadly identify those with early roles in neural crest development. To this end, we analyzed existing single-cell RNA-sequencing (scRNA-seq) data (*Williams et al., 2019*) from specification-stage avian embryos to identify enriched RNA-binding protein candidates. Using this approach, we identified *Elavl1* as an enriched transcript in newly formed neural crest cells. Elavl1 is a nucleocytoplasmic shuttling protein from the ELAV (embryonic lethal abnormal vision) family of RNA-binding proteins, which have conserved roles in neural development (*Ma et al., 1996*; *Yao et al., 1993*). It is a well-established stabilizer of mRNA, a function often mediated via association with the 3'-untranslated region (3'-UTR) of its mRNA targets (*Abdelmohsen and Gorospe, 2010*; *Rothamel et al., 2021*).

Elavl1 is essential for mammalian development and embryonic survival; Elavl1 null mouse embryos exhibit lethality due to abnormal placental morphogenesis, and conditional epiblast-null embryos display a broad array of phenotypes, ranging from defects in ossification and craniofacial development to asplenia. Interestingly, despite the myriad of tissue systems affected by Elavl1 knockout and relatively broad expression in wild-type embryos, mechanistic insights suggest Elavl1 acts on specific gene networks in a spatiotemporally controlled manner (*Katsanou et al., 2009*). Thus, due to its complexity and specificity of function, much remains to be discovered with

respect to Elavl1's essential roles and targets during embryonic development across tissue-specific contexts.

Here, we sought to determine the role of Elavl1 during cranial neural crest specification by taking advantage of the chick embryo model, an amniote system in which it is possible to perturb Elavl1 function with precise spatiotemporal control via unilateral knockdown. The results show that perturbation of Elavl1 led to premature neural crest delamination, as well as significant reduction in the expression of genes within the neural crest specification GRN. We find that these effects were mediated by loss of *Draxin*, a direct mRNA target for stabilization by Elavl1. Our data demonstrate a critical role for Elavl1, and RNA-binding protein-mediated post-transcriptional control, in the regulation of a critical neural crest specification module.

## Results

### The RNA-binding protein Elavl1/HuR is expressed in cranial neural crest

Cranial neural crest cells are indispensable for proper craniofacial development (*van Limborgh et al., 1983*; *Vega-Lopez et al., 2018*). Whereas transcription factors have been well-established critical regulators of neural crest development and craniofacial morphogenesis reviewed in *Gou et al., 2015*, growing evidence indicates an essential role for post-transcriptional regulation in these processes (*Cibi et al., 2019*; *Copeland and Simoes-Costa, 2020*; *Dennison et al., 2021*; *Forman et al., 2021*). To identify RNA-binding proteins with potential roles in cranial neural crest specification, we analyzed scRNA-seq data for cranial neural crest isolated from avian embryos at the 5–6 somite stage (*Williams et al., 2019*) we identified three distinct clusters (neural, premigratory [pNC], and delaminating/migratory [mNC]) among which genes associated with the gene ontology (GO) term 'binds to 3'-UTR' were differentially expressed (*Figure 1A–E*). To isolate potential positive regulators of neural crest specification, we then performed GO term analysis for 'stabilizes RNA' and 'regulates translation' among the identified RNA-binding proteins; only the *Elavl1* gene was associated with all three GO terms and abundantly expressed in the isolated cranial neural crest cells (*Figure 1F, G*).

Given that Elavl1 knockout mice often display defects in craniofacial structures (*Katsanou et al., 2009*), and the transcript appeared enriched in premigratory cranial neural crest cells (*Figure 1E*), we hypothesized a potential role for Elavl1 during cranial neural crest specification. To test this possibility, we first examined the expression pattern of Elavl1 in the developing chick embryo. Early in neurulation, when the neural plate border (NPB) is established within the rising neural folds, Elavl1 expression was detected in the anterior open neural tube and closing neural folds surrounding the anterior neuropore but absent from Pax7-expressing NPB cells (*Figure 2A*). As the neural tube closed, when neural crest specification is complete, Elavl1 expression became enriched throughout the neural tube and overlapped with Pax7 expression in premigratory cranial neural crest cells (*Figure 2B–D*). Following cranial neural crest EMT, Elavl1 remained expressed in the migratory neural crest cells, as well as throughout the brain and neural tube (*Figure 2E–G*). Thus, Elavl1 is expressed in specified, premigratory cranial neural crest cells following establishment of the NPB and is retained during the onset of EMT and in early migrating cranial neural crest cells.

### Elavl1 downregulation alters cranial neural crest specification and delamination

To determine what, if any, role Elavl1 has in cranial neural crest specification, we perturbed Elavl1 function in the early embryo using a translation-blocking antisense morpholino oligo (MO). We electroporated control or Elavl1 MOs bilaterally into gastrula stage chick embryos and analyzed neural crest specification using quantitative fluorescent hybridization chain reaction (HCR) to measure expression of markers of specified neural crest at HH9 (*Figure 3A, B*). Given Elavl1's association with Wnt signaling (*Kim et al., 2015*) and the essential roles Wnt signaling plays during early neural crest development (*Milet and Monsoro-Burq, 2012*; *Rabadán et al., 2016*; *Simões-Costa and Bronner, 2015a*; *Steventon and Mayor, 2012*; *Wu et al., 2003*; *Yanfeng et al., 2003*), we focused on the Wnt effector *Axud1*, its target and neural crest specifier *FoxD3*, and the Wnt antagonist *Draxin* (*Hutchins and Bronner, 2018*; *Hutchins and Bronner, 2019*; *Simões-Costa et al., 2015b*). Following Elavl1 knockdown (*Figure 3—figure supplement 1*; 61.4 ± 0.9% of the control side, p < 0.001, paired *t*-test, n = 5 embryos, 15 sections), we observed significant reduction in the levels of *Axud1* (*Figure 3C*; 73.7

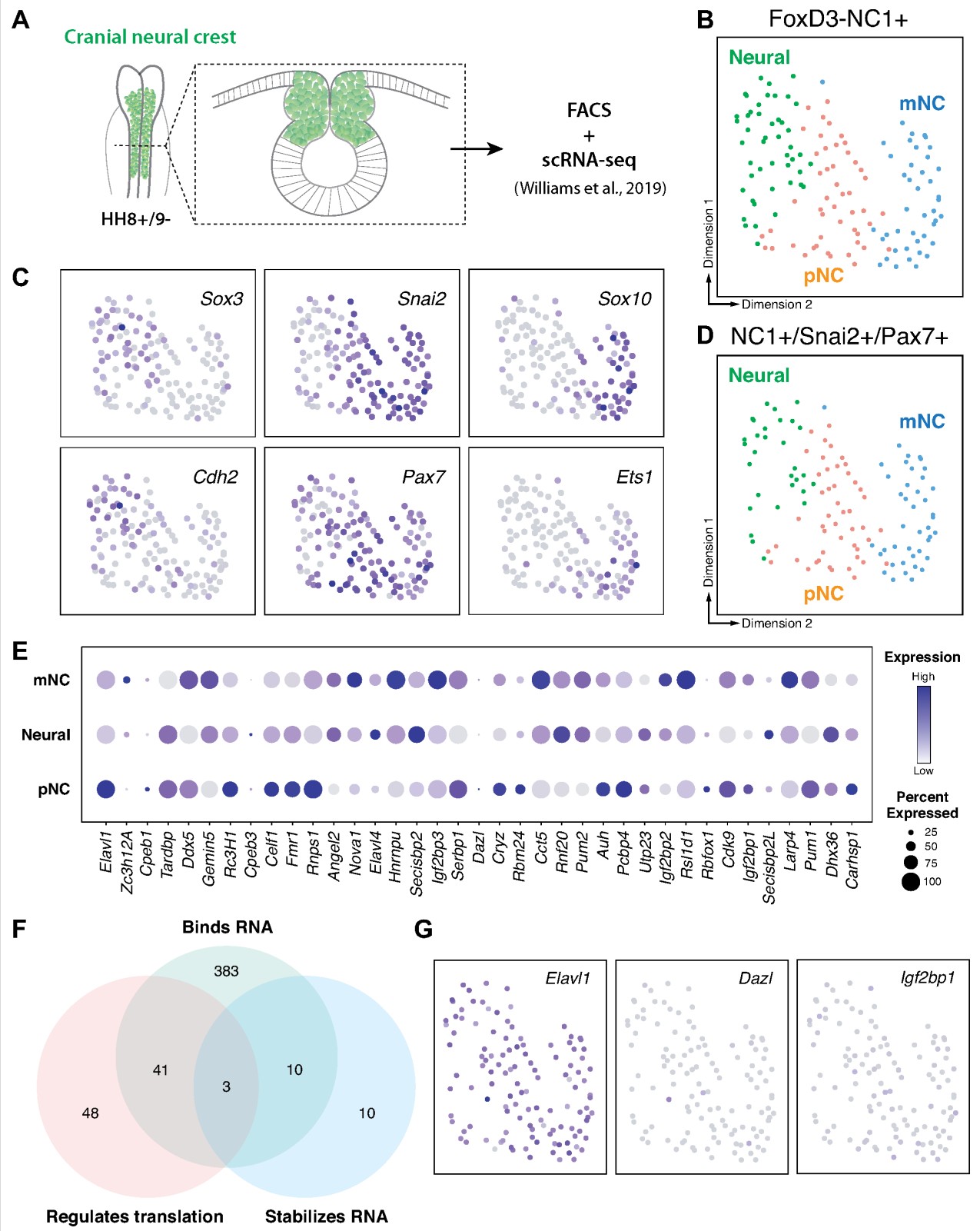

**Figure 1.** RNA-binding proteins are differentially expressed in premigratory and migratory cranial neural crest. (**A**) Schematic of early chick cranial neural crest cells at premigratory stages (HH8+/9−) in intact heads and cross-section expressing Citrine fluorescent protein under control of the FoxD3 NC1 enhancer used by Williams et al. to sort cranial neural crest cells for single-cell RNA-sequencing (scRNA-seq; *Williams et al., 2019*). (**B, C**) Dimensionality reduction using Uniform Manifold Approximation and Projection (UMAP) on published scRNA-seq data from *Williams et al., 2019*

*Figure 1 continued on next page*

*Figure 1 continued*

identified three distinct clusters neural, premigratory (pNC), and delaminating/migratory (mNC). Expression of marker genes (*Sox3* and *Cdh2* for neural, *Pax7* and *Snai2* for premigratory cranial neural crest, and *Sox10* and *Ets1* for early migratory cranial neural crest) was used to label the three subclusters. (**D**) The majority of the FoxD3-NC1+ cells were also positive for the expression of transcription factors *Pax7* and *Snai2*, which label the dorsal neural tube and premigratory/delaminating neural crest cells. These triple positive NC1+/Snai2+/Pax7+ were further processed for gene ontology analysis to identify post-transcriptional regulators. (**E**) A strip-plot showing expression and abundance of a subset of genes that are associated with the gene ontology term 'binds to 3'-UTR'. (**F**) A three-way Venn diagram shows overlap between genes associated with the gene ontology terms 'binds RNA', 'regulates translation', and 'stabilizes RNA'. Only three genes, *Elavl1*, *Dazl*, and *Igf2bp1*, were associated with all three. (**G**) Feature plots showing the expression distribution of the three genes identified in (**F**). Only *Elavl1* is abundant among all NC1+/Pax7+/Snai2+ cells.

± 3.1% of the control side, p = 0.03, Wilcoxon signed-rank test, *n* = 6 embryos), *FoxD3* (*Figure 3D*; 61.3 ± 5.5% of the control side, p = 0.002, Wilcoxon signed-rank test, *n* = 10 embryos), and *Draxin* (*Figure 3E*; 66.6 ± 1.8% of the control side, p = 0.03, Wilcoxon signed-rank test, *n* = 6 embryos) transcripts compared to contralateral control sides. To parse whether this was a specific effect or a broad defect in neural crest development, we also examined additional neural crest genes and found no significant difference in expression between control or Elavl1 knockdown sides for *Pax7* (*Figure 3F*; 87.4 ± 3.1% of the control side, p = 0.1, Wilcoxon signed-rank test, *n* = 5 embryos) or *Tfap2b* (*Figure 3G*; 98.2 ± 7.0% of the control side, p = 0.6, Wilcoxon signed-rank test, *n* = 4 embryos), indicating a specific defect in a subset of genes required for cranial neural crest specification.

We also performed immunostaining for Pax7 in cross-section to assess if Elavl1 knockdown altered neural crest cell number or dorsal neural tube morphology. Interestingly, the total number of Pax7 + cells was unaffected with Elavl1 knockdown (101.3 ± 4.5% of the control side, p = 0.6, one-sample Wilcoxon signed-rank test, *n* = 4 embryos, 11 sections); however, we found a significant increase in the number of Pax7 + cells that delaminated from the neural tube (139.4 ± 8.4% of the control side, p = 0.002, one-sample Wilcoxon signed-rank test), and concomitant decrease in the number of Pax7 + cells retained within the dorsal neural tube (66.0 ± 5.3% of the control side, p = 0.001, one-sample Wilcoxon signed-rank test: *Figure 3H, I*). To determine what impact premature delamination might have on cranial neural crest migration, we also performed whole mount immunostaining for Pax7 at HH9 + and found significant decrease in cranial neural crest emigration away from the midline (*Figure 3—figure supplement 2*; 63.6 ± 4.1% of the control side, p < 0.001, Wilcoxon matched-pairs signed-rank test, *n* = 5 embryos, 5 measurements per embryo averaged). Taken together, these data suggest that Elavl1 is required during early cranial neural crest development to regulate specification and prevent premature delamination.

### *Draxin* is the primary target of Elavl1 during cranial neural crest specification

To parse the mechanism of Elavl1 function during cranial neural crest specification, we sought to more broadly identify the RNA targets of Elavl1 using bulk RNA-sequencing (RNA-seq) and differential gene expression analysis (*Figure 4A*). Given that Elavl1 is known to bind to and stabilize its RNA targets via 3'-UTR interaction (*Chen et al., 2002*; *Dormoy-Raclet et al., 2007*; *Katsanou et al., 2009*; *Rothamel et al., 2021*; *Shi et al., 2020*), we expected expression of potential targets to be reduced with Elavl1 knockdown. Among the differentially expressed genes we identified (*Figure 4—figure supplement 1*), 12 were significantly downregulated, with four having established roles in neural crest development and which we validated using HCR—*Axud1*, *Draxin*, *BMP4*, and *Msx1* (*Figure 4B*, *Figure 4—figure supplement 1*). We also identified several canonical neural crest genes that were unaffected with Elavl1 knockdown (e.g., *Pax7*, *Tfap2b*, *Snai2*, *Sox9*, and *Zeb2*), consistent with HCR data (*Figure 3*, *Figure 4—figure supplement 1*). Together these data suggest Elavl1 does not broadly bind and stabilize the transcripts of neural crest genes, rather it targets specific RNAs to drive cranial neural crest specification.

Notably, *FoxD3* failed to meet our stringency cutoff during RNA-seq analysis due to low expression at the examined stages, and is likely downregulated with Elavl1 knockdown (*Figure 3D*) due to indirect effects from loss of *Axud1* (*Simões-Costa et al., 2015b*) and therefore unlikely to be a *bona fide* target of Elavl1. To determine whether Elavl1 directly or indirectly interacts with *Axud1*, *Draxin*, *BMP4*, and *Msx1* mRNAs, we first measured the rate of mRNA decay with actinomycin D treatment to assess the stability of these RNAs, and *Pax7* as a nontarget, with or without Elavl1 knockdown

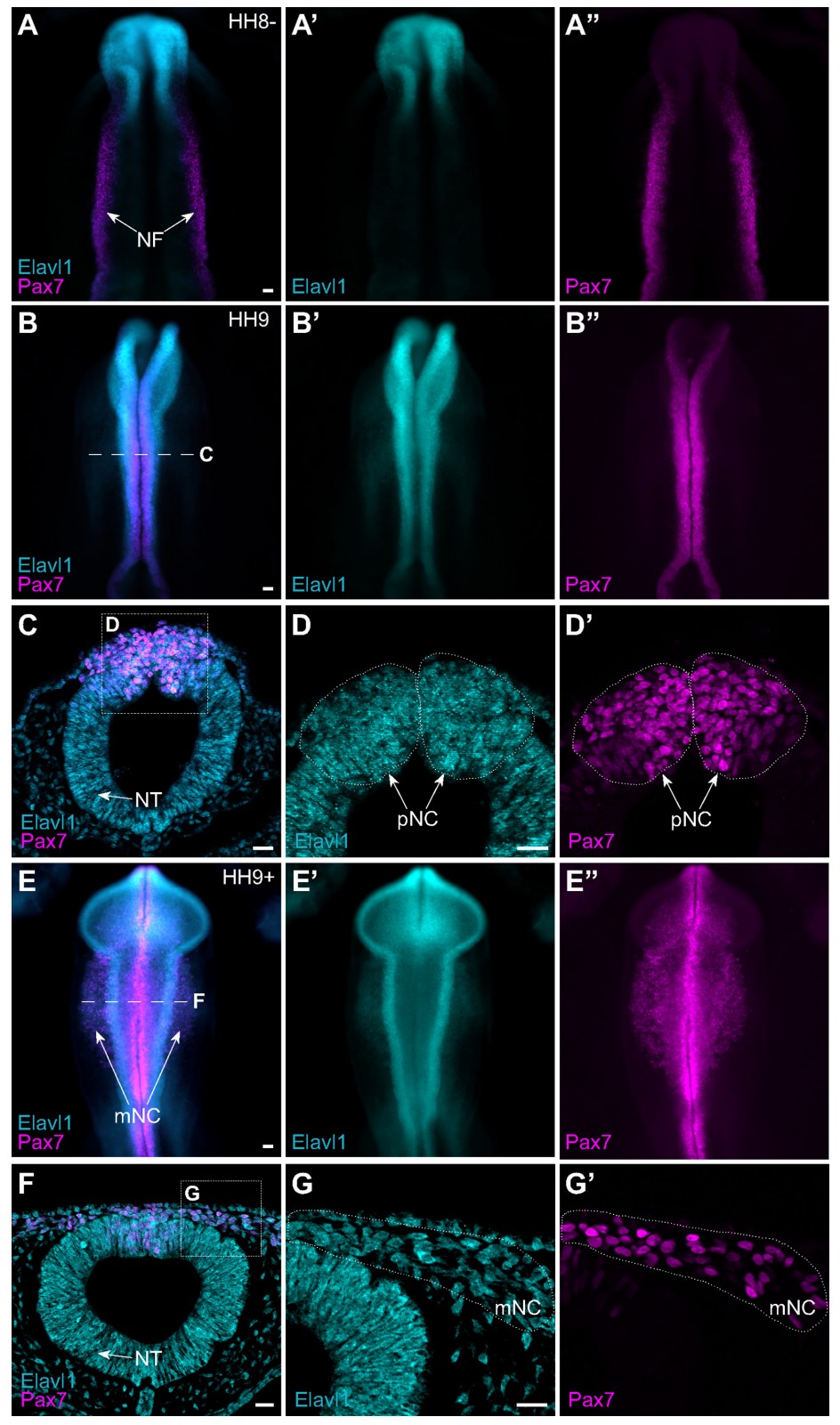

**Figure 2.** The RNA-binding protein Elavl1 is expressed in premigratory and migratory cranial neural crest. Representative epifluorescence images of wild-type HH8− (**A**), HH9 (**B–D**), and HH9+ (**E–G**) chick embryos, in whole mount (**A, B, E**) and cross-section (**C, D; F, G**) immunostained for Elavl1 (cyan) and Pax7 (magenta). Dashed white line (**B, E**) indicates level of cross-section (**C, D; F, G**), respectively; dotted white lines outline regions of

*Figure 2 continued on next page*

*Figure 2 continued*

premigratory and migratory neural crest as indicated. NF, neural folds; NT, neural tube; pNC, premigratory neural crest; mNC, migratory neural crest. Scale bar, 50 μm.

(*Figure 4C*); if Elavl1 is required for transcript stability, target RNAs should decay at a faster rate with loss of Elavl1 compared to control. Interestingly, among all the transcripts tested, only *Draxin* had a significant reduction in transcript stability (p = 0.001, Mann–Whitney test; *Figure 4D*), suggesting that *Draxin* is the primary target bound and stabilized by Elavl1 during cranial neural crest specification.

To test this hypothesis, we searched the 3'-UTR sequences of *Draxin*, *Axud1*, *BMP4*, *Msx1*, *Pax7*, and *Tfap2b* mRNAs for putative Elavl1-binding sites. The *Draxin* 3'-UTR contained four high probability-binding sites, whereas the other 3'-UTRs contained only one (*BMP4*, *Msx1*) or none (*Axud1*, *Pax7*, *Tfap2b*; *Figure 4—figure supplement 2*). Given that Elavl1 contains three RNA recognition motifs (RRMs) that cooperate for RNA recognition (*Pabis et al., 2019*), it is possible that multiple contacts are required for Elavl1 to bind and stabilize RNAs in vivo, and by extension is unlikely for Elavl1 to bind RNAs with only a single putative binding site. Thus, we hypothesize that Elavl1 specifically targets and stabilizes *Draxin* in cranial neural crest through multiple contact sites within its 3'-UTR.

To confirm that *Draxin* mRNA is bound by Elavl1 in vivo, we first performed an RNA immunoprecipitation (RIP) followed by quantitative reverse transcription-PCR (qRT-PCR) to pull down endogenous Elavl1 ribonucleoprotein (RNP) complexes. To this end, we incubated lysate generated from wild-type HH9 embryonic heads with magnetic beads coated with either Elavl1 antibody or a rabbit IgG nonspecific control antibody, then eluted bound RNA and performed qRT-PCR, comparing immunoprecipitated RNAs ('IP') with RNAs extracted from a fraction of the input lysate ('Input'). We expected that RNAs bound by Elavl1 would be enriched in the IP compared to the Input, whereas nonspecifically associated RNAs, that is nontargets, would not (*Figure 4E*). We found that *Pax7* and *FoxD3* mRNAs (nontargets) were neither enriched in the IP, nor significantly different from each other (p = 0.89, one-way analysis of variance (ANOVA) with Tukey's post hoc test); however, *Draxin* mRNA was significantly enriched with Elavl1 IP compared to *Pax7* and *FoxD3* (p < 0.001, one-way ANOVA with Tukey's post hoc test; *Figure 4F*), suggesting that Elavl1 specifically associated with endogenous *Draxin* mRNA. However, this assay could not distinguish whether *Draxin* was bound directly or indirectly, or where within the transcript it might be associating with Elavl1.

To test whether *Draxin* was directly bound by Elavl1 within the 3'-UTR, we performed a proximity ligation assay (PLA), wherein in situ fluorescent signal can be detected as puncta only when two proteins are in close proximity (<40 nm), indicating a direct interaction in vivo. Taking advantage of the MS2-MCP reporter system (*Tutucci et al., 2018*), we electroporated a construct encoding a GFP-tagged MS2 bacteriophage coat protein (MCP-GFP) alone ('Control') or in combination with ('Experimental') a construct containing a Luciferase coding region, MS2 stem loops (bound by MCP when transcribed), and the endogenous *Draxin* 3'-untranslated region (MS2-*Draxin* 3'-UTR) and performed PLA with antibodies against Elavl1 and GFP (*Figure 4G*). We observed significantly more PLA puncta with expression of MS2-*Draxin* 3'-UTR (*Figure 4H–I*), indicating a specific and direct interaction between Elavl1 and the *Draxin* 3'-UTR.

## Elavl1 maintains cranial neural crest specification via *Draxin* mRNA stabilization

Our data suggest that Elavl1 specifically binds and stabilizes *Draxin* mRNA as its primary target during cranial neural crest specification. To determine if defects in cranial neural crest specification with Elavl1 knockdown were indirect, and due to *Draxin* downregulation, we examined *FoxD3*, *Axud1*, *Msx1*, and *BMP4* expression in Draxin knockdown embryos. We electroporated control and Draxin MO (*Hutchins and Bronner, 2018*; *Hutchins and Bronner, 2019*) bilaterally, and performed HCR at HH9. As with Elavl1 MO (*Figure 3*), we observed significant reduction in the levels of *FoxD3* (*Figure 5A*; 26.8 ± 3.2% of the control side, p < 0.02, Wilcoxon matched-pairs signed rank, *n* = 7 embryos), *Axud1* (*Figure 5B*; 48.8 ± 5.1% of the control side, p < 0.01, paired *t*-test, *n* = 7 embryos), *Msx1* (*Figure 5C*; 49.7 ± 5.8% of the control side, p < 0.01, paired *t*-test, *n* = 4 embryos), and *BMP4* (*Figure 5D*; 63.6 ± 4.0% of the control side, p < 0.01, paired *t*-test, *n* = 4 embryos) transcripts compared to contralateral control sides (*Figure 5E*).

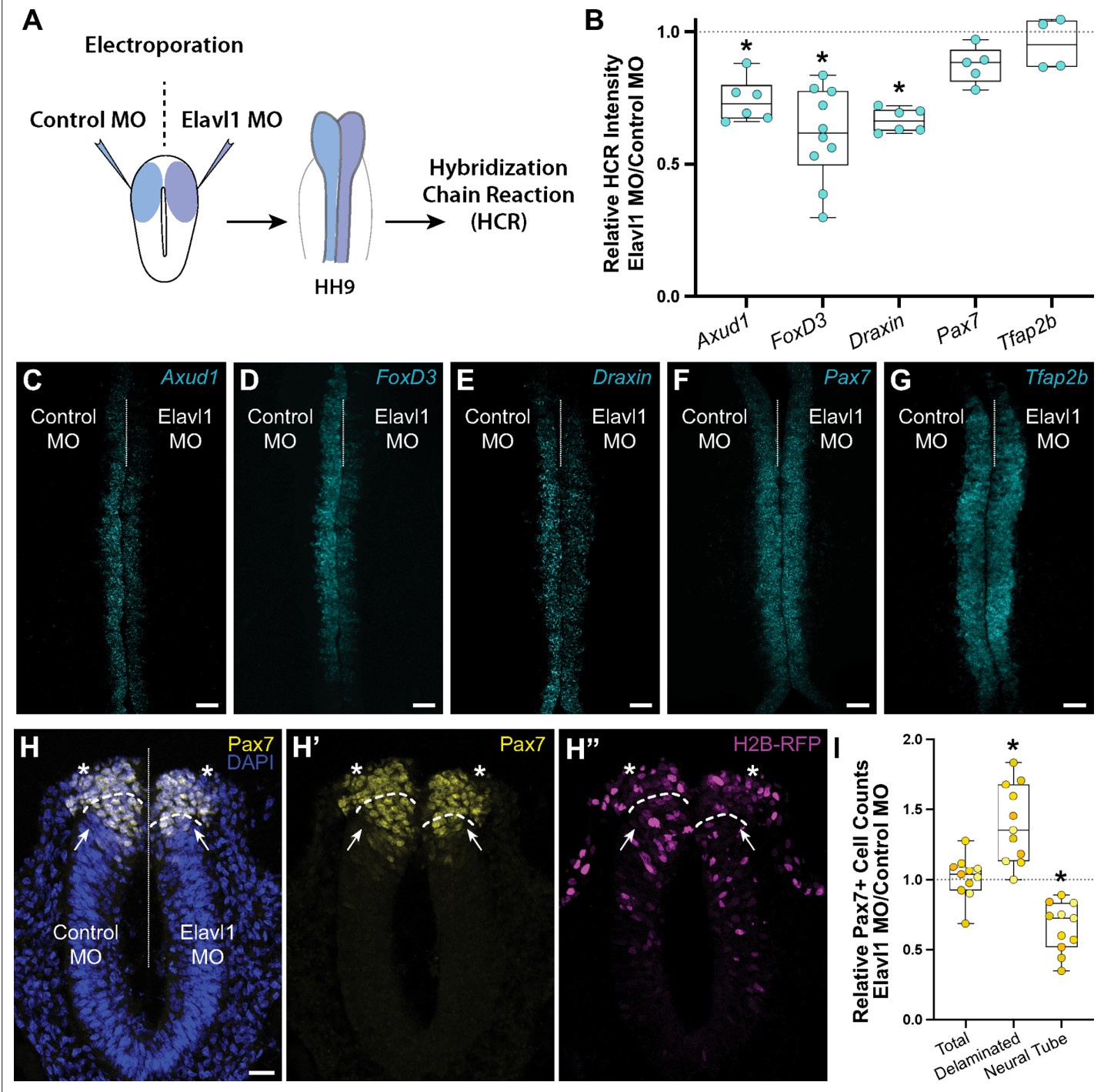

**Figure 3.** Elavl1 knockdown alters cranial neural crest specification and delamination. (**A**) Schematic diagram illustrating experimental design. Gastrula stage chick embryos were electroporated bilaterally with a standard control and translation-blocking morpholino (MO) targeting *Elavl1*. Electroporated embryos were subsequently processed for quantitative hybridization chain reaction (HCR) and analyzed in whole mount, comparing the knockdown to the contralateral control side. (**B**) Quantitation of HCR processed embryos for control versus Elavl1 knockdown for cranial neural crest transcripts, calculated as ratio of Elavl1 MO versus control MO integrated density. Representative confocal maximum intensity projection micrographs for *Axud1* (*n* = 6) (**C**), *FoxD3* (*n* = 10) (**D**), *Draxin* (*n* = 6) (**E**), *Pax7* (*n* = 5) (**F**), and *Tfap2b* (*n* = 4) (**G**) transcripts. Dotted white line indicates midline. MO, morpholino. Scale bar, 50 µm. *p < 0.05, Wilcoxon signed-rank test. (**H**) Representative apotome maximum intensity projection micrographs of cross-sectioned embryo bilaterally co-electroporated with a fluorescent electroporation control construct (H2B-RFP) and control MO (left) or Elavl1 MO (right) immunostained for Pax7 (yellow). Nuclei were stained with DAPI (4',6-diamidino-2-phenylindole)(blue). Dotted white line indicates midline. Dashed white lines indicate limit of dorsal neural tube. Arrows indicate 'neural tube' Pax7 cells. Asterisks indicate 'delaminated' Pax7 cells. Scale bar, 20 µm. (**I**)

*Figure 3 continued on next page*

*Figure 3 continued*

Quantification of the ratio of Pax7 + cells on Elavl1 MO (right) versus control MO (left) sides of cross-sections. Data are from individual sections; sections from same embryo are displayed in same color (*n* = 4 embryos, 11 sections). *p ≤ 0.002, one-sample Wilcoxon signed-rank test.

The online version of this article includes the following figure supplement(s) for figure 3:

**Figure supplement 1.** Translation-blocking morpholino suppresses Elavl1 expression.

**Figure supplement 2.** Elavl1 knockdown inhibits cranial neural crest emigration.

We next asked whether *Draxin* upregulation alone was sufficient to rescue the Elavl1 MO phenotype (*Figure 3B–E*). To this end, we co-electroporated Elavl1 MO with a Draxin overexpression construct (Draxin-FLAG; *Hutchins and Bronner, 2018*; *Hutchins and Bronner, 2019*), and assessed neural crest specification with HCR. Indeed, exogenous *Draxin* was sufficient to significantly restore *FoxD3* (*Figure 5F*; 84.5 ± 9.6% of the control side, *n* = 6 embryos), *Axud1* (*Figure 5G*; 90.3 ± 5.0% of the control side, *n* = 6 embryos), *Msx1* (*Figure 5H*; 91.0 ± 1.5% of the control side, *n* = 5 embryos), and *BMP4* (*Figure 5I*; 96.5 ± 3.1% of the control side, *n* = 5 embryos) expression from Elavl1 knockdown (p < 0.05, one-tailed paired *t*-test) to expression levels not significantly different from control (*Figure 5J*; p > 0.09, Wilcoxon signed-rank test). To determine if exogenous *Draxin* was sufficient to rescue the premature delamination phenotype (*Figure 3H, I*) caused by Elavl1 knockdown, we also performed immunostaining for Pax7 in cross-section to assess neural crest cell number and dorsal neural tube morphology. Indeed, co-electroporation of Draxin-FLAG with Elavl1 MO was able to rescue the number of Pax7 + cells that delaminated and remained within the dorsal neural tube to near-control levels (*Figure 5L*; 91.6 ± 4.0% and 105.8 ± 12.0% of the control side, respectively, *n* = 3 embryos, 6 sections, p > 0.12, one-sample Wilcoxon signed-rank test). Taken together, these data indicate that *Draxin* mRNA is the primary target of Elavl1 in premigratory cranial neural crest and is stabilized via 3'-UTR interaction to maintain neural crest specification.

## Discussion

Understanding of neural crest development has been greatly enhanced by the identification of key transcriptional circuits that control its developmental progression. Recent studies suggest a critical role for post-transcriptional regulation in the refinement of the expression outputs of these GRNs. Here, we identified and characterized Elavl1 as an RNA-binding protein essential for the maintenance of cranial neural crest specification via its stabilization of the Wnt antagonist *Draxin*. We found that loss of Elavl1, and by extension its target, *Draxin*, interferes with output from multiple nodes of the GRNs required for neural crest specification, including those driven by Wnt and BMP (*Hovland et al., 2020*; *Simões-Costa et al., 2015b*; *Tribulo et al., 2003*). Together, our data implicate Elavl1 as a point of integration to coordinate signaling from parallel but independent GRNs.

At a mechanistic level, our study is consistent with previous work examining Elavl1 function in other cellular contexts, with respect to its role as a stabilizing RNA-binding protein via 3'-UTR interaction (*Chen et al., 2002*; *Dormoy-Raclet et al., 2007*; *Katsanou et al., 2009*; *Rothamel et al., 2021*; *Shi et al., 2020*). In the context of embryonic development, our study aligns well with data from knockout mouse consistent with an important role for Elavl1 in intersecting signaling cascades; interestingly, this prior work similarly observed indirect downregulation of *BMP4* with loss of Elavl1, though an upstream mediator remained unidentified (*Katsanou et al., 2009*). In neural crest specification, we identified a single biologically relevant target of Elavl1, whereas prior high-throughput studies found many RNAs directly bound by Elavl1 (*Mukherjee et al., 2011*; *Rothamel et al., 2021*). Whether this is a feature specific to neural crest is unclear, though the knockout mouse work suggests that during embryonic development Elavl1 function and RNA targets are driven by spatiotemporal determinants (*Katsanou et al., 2009*), which may be due RNP complex heterogeneity as a result of tissue-specific expression of other interacting RNA-binding proteins. Taken in the context of these prior studies, we speculate that despite broad tissue expression of Elavl1 across embryonic development, specificity in neural crest is achieved through spatiotemporally regulated combinatorial expression of post-transcriptional regulators and RNA regulons (*Keene, 2007*; *Keene and Tenenbaum, 2002*).

It is important to note that, while Elavl1 expression persists in cranial neural crest during the initiation of EMT and migration, *Draxin* must be rapidly downregulated for these processes to proceed

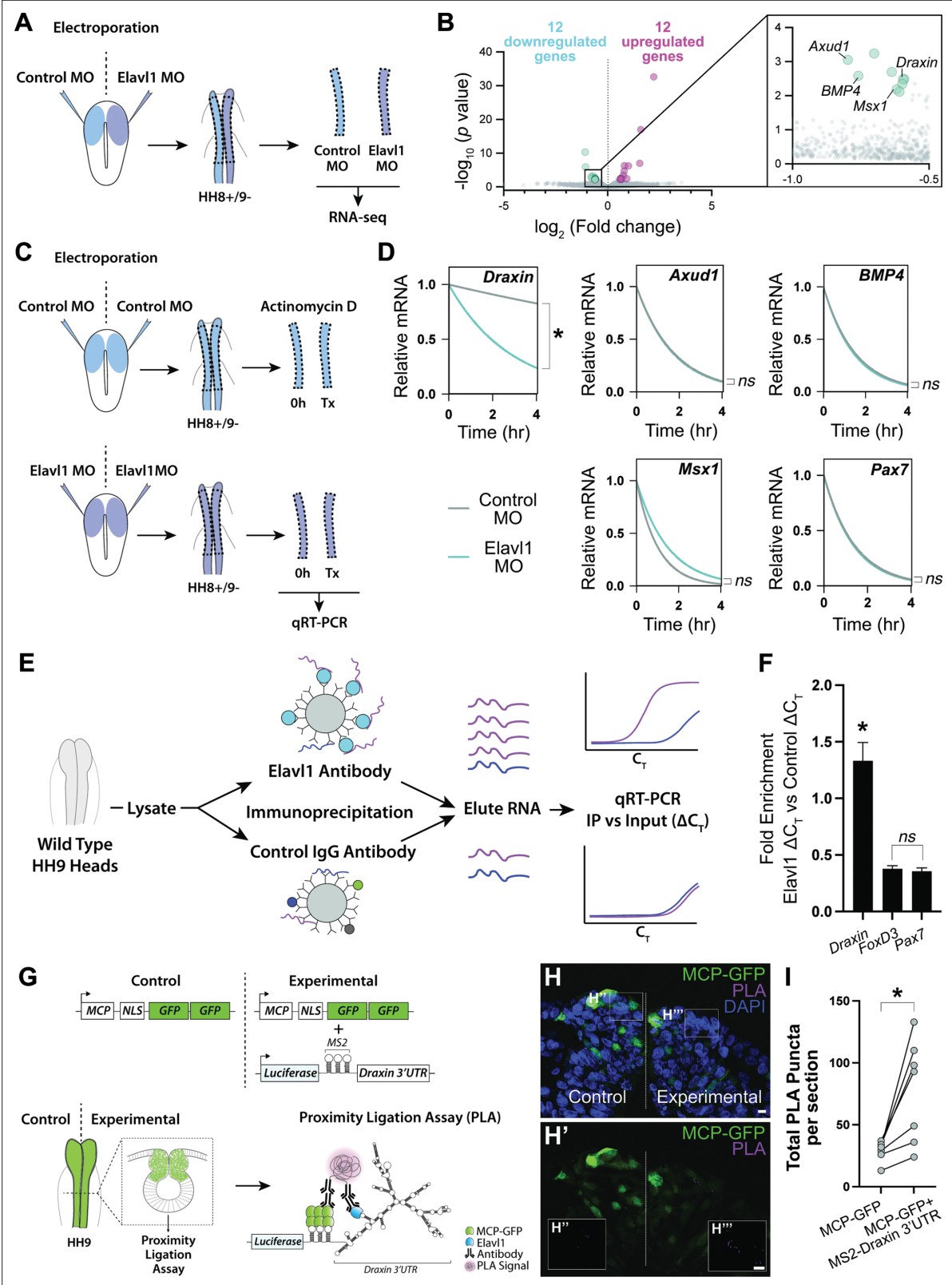

**Figure 4.** *Draxin* mRNA is the primary target of Elavl1 during cranial neural crest specification. (**A**) Schematic diagram illustrating experimental design for RNA-sequencing (RNA-seq). Gastrula stage chick embryos were electroporated bilaterally with a standard control and translation-blocking morpholino (MO) targeting *Elavl1*. Dorsal neural folds were dissected from stage HH8+/9– embryos, pooled (*n* = 3), and processed for bulk RNA-seq (three biological replicates). (**B**) Volcano plot following differential expression analysis and filtering of RNA-seq data. Of the 24 genes differentially

*Figure 4 continued on next page*

*Figure 4 continued*

expressed following Elavl1 knockdown (12 upregulated, 12 downregulated), four genes (*Draxin, Axud1, Msx1, BMP4*) have established roles in neural crest development and were significantly downregulated. (**C**) Schematic diagram illustrating experimental design for RNA stability assay. Gastrula stage chick embryos were electroporated bilaterally with control or translation-blocking morpholino (MO) targeting *Elavl1*. Dorsal neural folds were dissected from stage HH8+/9− embryos; left neural folds were used as the 0 hr time point, whereas right neural folds were treated with actinomycin D for 30 min, 2 hr, or 4 hr prior to total RNA extraction and quantitative reverse transcription-PCR (qRT-PCR) to measure RNA decay. (**D**) Transcript stability plots show *Draxin* mRNA stability is significantly reduced (*p = 0.001, Mann–Whitney test) with Elavl1 knockdown (blue) compared to control (gray), whereas other neural crest mRNAs (*Axud1, Msx1, BMP4, Pax7*) are not (*ns*, nonsignificant, p > 0.37, Mann–Whitney test). (**E**) Schematic illustrating experimental design of RNA-binding protein/RNA co-immunoprecipitation (RIP) to test RNA association with Elavl1 in vivo for neural crest targets. Lysates generated from HH9 heads were incubated with antibody-coated beads for Elavl1 or a nonspecific IgG to co-immunoprecipitate protein with bound RNAs. In qRT-PCR, specifically bound RNAs would be more abundant and reach threshold before RNAs that were nonspecific, and therefore would have smaller $C_T$ values. $C_T$, threshold cycle. (**F**) Fold enrichment of RNAs eluted from RIP (n = 16 embryos), quantified by qRT-PCR, performed in triplicate. *ns*, nonsignificant, p = 0.89, one-way analysis of variance (ANOVA) with Tukey's post hoc test. *p < 0.001, one-way ANOVA with Tukey's post hoc test. Error bars, standard error of the mean (SEM). (**G**) Schematic diagram illustrating experimental design for proximity ligation assay (PLA). Gastrula stage chick embryos were electroporated bilaterally with a construct expressing a nuclear localized, GFP-tagged MS2 bacteriophage coat protein (MCP-GFP) alone (left) or in combination with a construct containing a Luciferase coding region, MS2 stem loops (which are bound by MCP when transcribed), and the endogenous *Draxin* 3'-untranslated region (MS2-*Draxin* 3'UTR) (right). Following fixation and cross-sectioning at HH9, tissues were incubated with primary antibodies made in goat and rabbit that recognized GFP and Elavl1, respectively. Secondary antibodies against goat and rabbit IgG were labeled with complementary oligonucleotides that generate a fluorescent signal due to rolling circle amplification only when in close proximity (<40 nm). Thus, fluorescence signal (magenta) would indicate in vivo interaction between MCP-GFP and endogenous Elavl1. (**H**) Representative confocal maximum intensity projection micrograph of dorsal neural folds from cross-sectioned HH9 embryo bilaterally electroporated with MCP-GFP (green) alone ('control', left) or with MS2-*Draxin* 3'-UTR ('experimental', right), processed for PLA (magenta) as illustrated in panel (**G**), and stained for DAPI (blue). Boxes in (**H**) indicate zoomed-in areas in (**H″**) and (**H‴**). Scale bar, 5 μm. (**I**) Quantitation of total number of PLA puncta per section for (n = 3 embryos, 2 sections/embryo). *p = 0.016, two-tailed Wilcoxon matched-pairs signed-rank test.

The online version of this article includes the following figure supplement(s) for figure 4:

**Figure supplement 1.** RNA-sequencing (RNA-seq) identified four neural crest genes specifically downregulated with Elavl1 knockdown.

**Figure supplement 2.** Putative Elavl1-binding sites within the *Draxin* 3'-untranslated region (UTR) predicts a direct interaction.

---

(*Hutchins and Bronner, 2018*; *Hutchins and Bronner, 2019*; *Hutchins et al., 2021*). Thus, we hypothesize that Elavl1 becomes endogenously displaced from *Draxin* at the onset of EMT, though it is yet unclear how this is achieved. RNA-binding proteins are known to alter association with targets due to post-translational modifications such as phosphorylation or alternative RNA-binding protein competition (*Dassi, 2017*; *García-Mauriño et al., 2017*; *Liu et al., 2015*). Indeed, inhibition of serine–threonine kinases has been shown in neural crest to increase cell–cell adhesions and negatively impact cell migration (*Monier-Gavelle and Duband, 1995*), suggesting kinase-driven signaling pathway activation coincident with neural crest EMT; this is interesting given established roles for serine–threonine phosphorylation in modulating Elavl1-RNA-binding activity and target selection (*Grammatikakis et al., 2017*). Thus, we speculate that Elavl1 serine–threonine phosphorylation, either in combination with or as an alternative to competitive RNA-binding protein displacement, likely facilitates *Draxin* release and turnover at the onset of EMT.

Because its primary target during specification is downregulated while Elavl1 expression persists, this suggests there are likely additional targets and roles for Elavl1 beyond *Draxin* stabilization. Elavl1 has been shown in other contexts to stabilize Snail1 (*Dong et al., 2007*) and matrix metalloprotease-9 (MMP-9) (*Yuan et al., 2011*), factors with well-established roles in neural crest EMT (*Cano et al., 2000*; *Kalev-Altman et al., 2020*; *Monsonego-Ornan et al., 2012*; *Strobl-Mazzulla and Bronner, 2012*; *Taneyhill et al., 2007*). Given that the neural crest specification GRNs is proceeded by activation of an EMT GRN also coinciding with Elavl1 expression, we speculate that Elavl1 likely intersects with additional GRNs following specification. It is possible one or more of the downregulated genes we identified through RNA-seq that were not important for specification may yet be targets of Elavl1, with functions later in neural crest development. To fully understand how Elavl1 exerts control over key developmental processes, these possibilities will need to be explored.

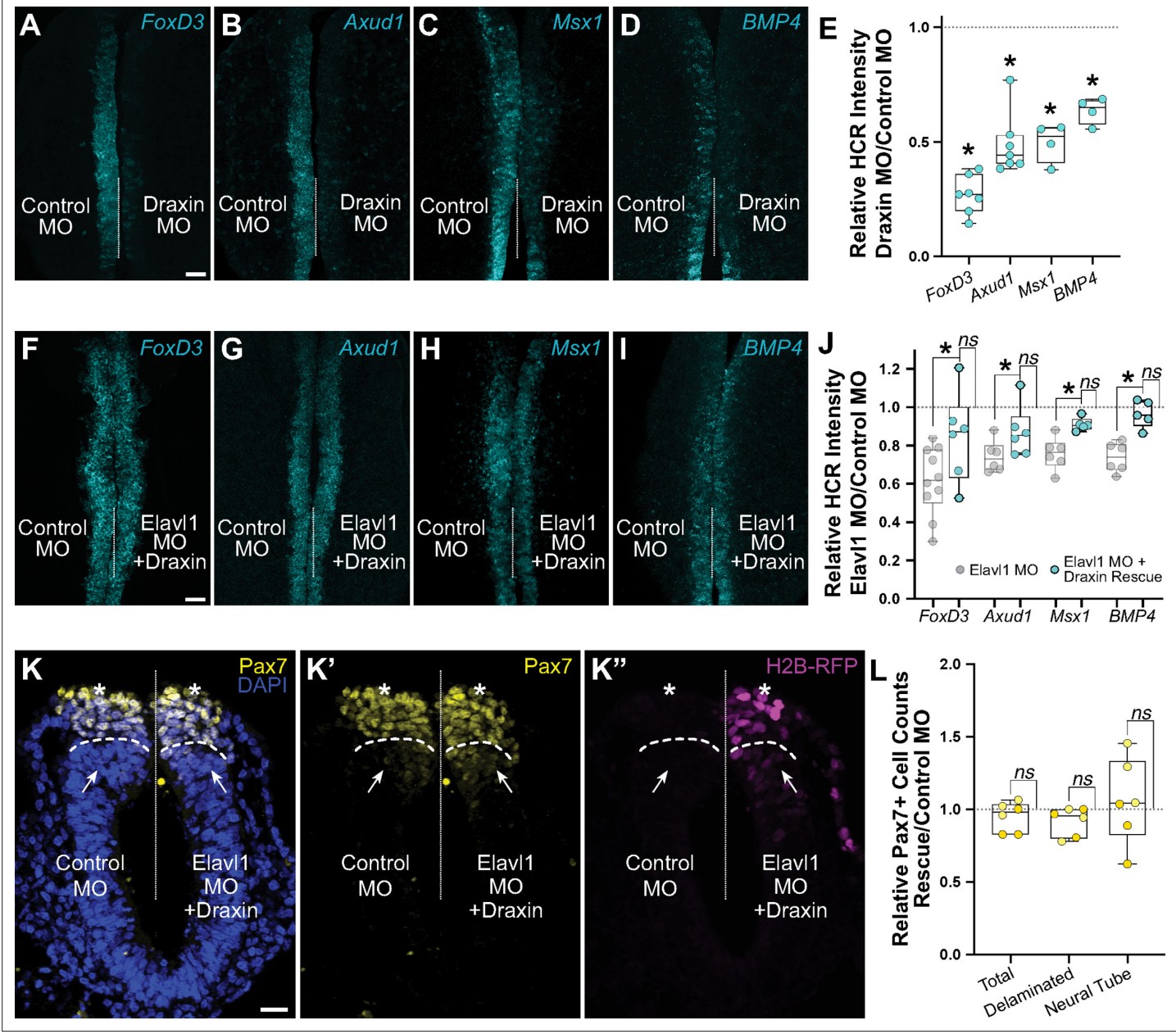

**Figure 5.** Elavl1 knockdown phenotypes are rescued by *Draxin*. Representative confocal maximum intensity projection micrographs of hybridization chain reaction (HCR) processed embryos for *FoxD3* (*n* = 7) (**A**), *Axud1* (*n* = 7) (**B**), *Msx1* (*n* = 4) (**C**), and *BMP4* (*n* = 4) (**D**) transcripts in whole mount embryos bilaterally electroporated with control morpholino (left) or Draxin morpholino (right). Dotted white line indicates midline. MO, morpholino. Scale bar, 50 μm. (**E**) Quantitation of HCR processed embryos for control versus Elavl1 knockdown, calculated as ratio of Elavl1 MO versus control MO integrated density. *p < 0.02, Wilcoxon matched-pairs signed rank or paired *t*-test as indicated in text. Representative confocal maximum intensity projection micrographs of HCR processed embryos for *FoxD3* (*n* = 6) (**F**), *Axud1* (*n* = 6) (**G**), *Msx1* (*n* = 5) (**H**), and *BMP4* (*n* = 5) (**I**) transcripts in whole mount embryos bilaterally electroporated with control morpholino (left) or Elavl1 morpholino and a Draxin overexpression construct (right). Dotted white line indicates midline. MO, morpholino. Scale bar, 50 μm. (**J**) Quantitation of HCR processed embryos for control versus Elavl1 knockdown with Draxin rescue, calculated as ratio of Elavl1 MO + Draxin versus control MO integrated density. *ns*, nonsignificant, p > 0.09, Wilcoxon signed-rank test. Grayed values indicating Elavl1 MO data were originally presented in *Figure 3* and are reproduced here to illustrate rescue. *p < 0.05, one-tailed paired *t*-test. (**K**) Representative apotome maximum intensity projection micrographs of cross-sectioned embryo bilaterally electroporated with control MO (left) or Elavl1 MO + Draxin overexpression (right), immunostained for Pax7 (yellow). Draxin overexpression is indicated by expression of H2B-RFP (magenta). Nuclei were stained with DAPI (blue). Dotted white line indicates midline. Dashed white lines indicate limit of dorsal neural tube. Arrows indicate 'neural tube' Pax7 cells. Asterisks indicate 'delaminated' Pax7 cells. Scale bar, 20 μm. (**L**) Quantification of the ratio of Pax7 + cells on Elavl1 MO + Draxin (right, 'Rescue') versus control MO (left) sides of cross-sections. Data are from individual sections; sections from same embryo are displayed in same color (*n* = 3 embryos, 6 sections). *ns*, nonsignificant, p > 0.12, one-sample Wilcoxon signed-rank test.

## Materials and methods

### Model organism and embryo collection

Fertile chicken eggs (*Gallus gallus*) were purchased locally (Sun State Ranch, Monrovia, CA), and incubated in a humidified 37°C incubator to the specified Hamburger–Hamilton (HH) stage (*Hamburger and Hamilton, 1951*). Live embryos were removed from eggs with Whatman filter paper as described (*Hutchins and Bronner, 2018*; *Hutchins and Bronner, 2019*) and stored in Ringer's solution until further processing.

### Immunohistochemistry and HCR

For whole mount immunohistochemistry, embryos were fixed at room temperature for 20 min with 4% paraformaldehyde in sodium phosphate buffer. For cross-sections, embryos were fixed at room temperature for 1 hr, then washed, embedded, and cryosectioned as described (*Hutchins and Bronner, 2018*; *Hutchins and Bronner, 2019*) prior to immunohistochemistry. Washes, blocking (10% donkey serum), and antibody incubations were performed in TBSTx (0.5 M Tris–HCl/1.5 M NaCl/10 mM $CaCl_2$/0.5% Triton X-100/0.001% Thimerosal) as described (*Chacon and Rogers, 2019*; *Manohar et al., 2020*). Primary antibodies are listed in the Key Resources Table. Species-specific secondary antibodies were labeled with Alexa Fluor 568 and 647 (Invitrogen) and used at 1:1000 or 1:500, respectively. For nuclear staining on cross-sections, DAPI (4′,6-diamidino-2-phenylindole) was added to the secondary antibody solution at [14.3 µM] final concentration. Coverslips were mounted using Fluoromount-G (SouthernBiotech).

HCR was performed as described (*Gandhi et al., 2020*). Embryos were fixed at room temperature for 1 hr with 4% paraformaldehyde in phosphate-buffered saline (PBS) prior to HCR processing. Custom HCR probes were designed and ordered through Molecular Technologies.

### Gene expression constructs and perturbation

Translation-blocking antisense MO for Elavl1 (Gene Tools; Key Resources Table) was designed to span the *Elavl1* (GenBank: NM_204833.1) start codon from nucleotide −20 to +5, and electroporated at [2 mM]. Draxin MO was described previously (*Hutchins and Bronner, 2018*; *Hutchins and Bronner, 2019*), and electroporated at [1 mM]. The standard control MO (Gene Tools) was used for contralateral control electroporation. MOs were co-electroporated with pCIG (*Megason and McMahon, 2002*) or pCI-H2B-RFP (*Betancur et al., 2010*) to increase electroporation efficiency and to visualize successfully electroporated cells. The *Draxin*-FLAG overexpression construct (*Draxin* OE; *Hutchins and Bronner, 2018*; *Hutchins and Bronner, 2019*) and the MCP-GFP construct *Hutchins et al., 2020* have been previously described. The MS2-*Draxin* 3′-UTR construct used in this study was generated by exchanging the mTurq2 coding region for a Luciferase coding region in a previously described MS2-*Draxin* 3′-UTR construct (*Hutchins et al., 2020*). Electroporations were performed on HH4 gastrula stage chicken embryos as described previously (*Hutchins and Bronner, 2018*; *Hutchins and Bronner, 2019*).

### RNA-seq and data analysis

For single-cell RNA-seq analysis, raw data from *Williams et al., 2019* were downloaded from Gene Expression Omnibus (GEO; GSE130500) and processed for quality assessment using FastQC (*Andrews, 2014*). The reads were trimmed and filtered using Cutadapt (*Martin, 2011*), following which they were aligned to the chicken galgal6 (GRCg6a) genome assembly using Bowtie2 (*Langmead and Salzberg, 2012*). The alignment files generated were used to count features using HTSeq-Count (*Anders et al., 2015*). All further analyses were performed in R using Seurat (*Butler et al., 2018*) as previously described (*Gandhi et al., 2020*). GO terms associated with all genes in the chicken genome were identified using the biomaRt package (*Durinck et al., 2009*). This dataset was queried for the ontology terms expected to be associated with RNA-binding proteins, such as 'binds mRNA', 'binds 3′-UTR', 'regulates translation', and 'stabilizes RNA'. All feature plots and strip plots were made using Seurat.

For bulk RNA-seq, dorsal neural folds from morpholino-electroporated embryos grown to stage HH8+/9− were dissected in DEPC-treated PBS and pooled (*n* = 3) before total RNA was extracted using an RNAqueous-Micro kit (Thermo). cDNA libraries (three biological replicates per treatment) were prepared from 10 ng total RNA per pooled sample using the Takara Bio SMART-Seq v4 Ultra

Low Input cDNA kit, according to the manufacturer's instructions. Single-end RNA-sequencing was performed at the Caltech Millard and Muriel Jacobs Genetics and Genomics Laboratory at a depth of 20 million reads. Reads were processed and differential expression analysis was performed as described (*Hutchins et al., 2021*; *Piacentino et al., 2021*). We subsequently filtered the gene lists to exclude lowly expressed genes (average normalized count values <250), resulting in a filtered list of 24 differentially expressed genes (12 downregulated, 12 upregulated) with greater than 1.5-fold change and p < 0.05 cutoff.

## RNA decay assay

Embryos were electroporated bilaterally with either control MO or Elavl1 MO and grown to HH8+/9−. Dorsal neural folds were dissected in DEPC-treated PBS, pooled (*n* = 3), and suspended in explant culture media (Dulbecco's modified Eagle medium supplemented with 10% fetal bovine serum, 10% chick embryo extract, and 1% penicillin/streptomycin). Actinomycin D [10 µg/ml] or DMSO (dimethyl sulfoxide) alone (for 0 hr time point) was added to neural fold samples, which were then incubated at 37°C/5% $CO_2$ for 0 hr, 30 min, 2 hr, or 4 hr. Following the specified incubation time, neural fold samples were centrifuged at 300 × *g* for 3 min to remove media then washed three times with DEPC-treated PBS.

Total RNA was then extracted using the RNAqueous-Micro kit (Thermo) and 50 ng of RNA was reverse transcribed using SuperScript III and oligo dT priming. Following reverse transcription, we performed qPCR using FastStart Universal SYBR Green Master (Rox) with cDNA (diluted 1:10) and gene-specific primers (Key Resources Table) on a QuantStudio 3 Real-Time PCR System (Applied Biosystems) in triplicate. Average $C_T$ values were calculated for 0 hr time points. Individual $C_T$ values for 30 min, 2 hr, and 4 hr actinomycin D time points were subtracted from the 0 hr average $C_T$, and fold change was calculated for each technical replicate from the start of transcription inhibition (*t* = 0 hr) to each actinomycin D time point for each target. RNA decay rates were calculated and plotted using one phase decay in Prism9 (*n* = three biological replicates).

## RIP and qRT-PCR

RIP was performed as described (*Hutchins and Szaro, 2013*; *Jayaseelan et al., 2014*), with minor modifications. Briefly, Protein-G Dynabeads were washed with NT-2 (50 mM Tris–HCl, 150 mM NaCl, 1 mM $MgCl_2$, 0.05% NP-40), blocked in NT-2/5% bovine serum albumin (BSA) for 1 hr at room temperature, then incubated with 5 µg antibody (Elavl1 IgG or Control IgG) in NT-2/5% BSA for 1 hr at room temperature. Following antibody incubation, antibody-coated beads were washed with NT-2 and resuspended in NET-2 (NT-2, 20 mM EDTA (ethylenediaminetetraacetic acid), 400 U RNaseOUT, 1× cOmplete, Mini EDTA-free Protease Inhibitor) until addition of cleared lysate.

Embryonic heads were dissected in Ringer's solution, washed in RNase-free PBS, and dissociated in Accumax (Innovative Cell Technologies) for 15 min at room temperature. Following dissociation, cells were pelleted at 2000 × *g* for 4 min at 4°C, washed in RNase-free PBS, and resuspended in polysome lysis buffer (0.1 M KCl, 5 mM $MgCl_2$, 10 mM HEPES (4-(2-hydroxyethyl)-1-piperazineethanesulfonic acid), 0.5% NP-40, 200 U RNaseOUT, 1× cOmplete, Mini EDTA-free Protease Inhibitor). Cells were frozen at −80 °C overnight to complete lysis and reduce adventitious binding. Lysate was then thawed on ice, vortexed, and centrifuged at 14,000 × *g* for 10 min at 4°C to remove cellular debris, then cleared lysate was added to antibody-coated beads in NET-2. Immediately following addition of cleared lysate, 10% was removed to serve as an Input control. IP reaction was tumbled at room temperature for 1 hr, beads were then washed in NT-2, and RNA was eluted in proteinase K buffer (NT-2, 1% sodium dodecyl sulfate, 1.2 mg/ml proteinase K) for 30 min at 55°C and phenol/chloroform extracted.

RNA from Input and IP samples was reverse transcribed using SuperScript III and oligo dT priming. Following reverse transcription, we performed qPCR using FastStart Universal SYBR Green Master (Rox) with cDNA (diluted 1:5) and gene-specific primers (Key Resources Table) on a QuantStudio 3 Real-Time PCR System (Applied Biosystems) in triplicate. We determined $\Delta C_T$ ($\Delta C_T$ = Input $C_T$ − IP $C_T$) for *Draxin*, *FoxD3*, and *Pax7* for Elavl1 and Control IgG RIP samples, then calculated fold enrichment values (=$2^{(\text{Average Control IgG }\Delta CT-\text{ Elavl1 }\Delta CT)}$) for each target and replicate.

## Proximity ligation assay

Embryos were electroporated bilaterally with MCP-GFP alone ('control'), or MCP-GFP and MS2-*Draxin* 3'-UTR ('experimental'), grown to stage HH9, then fixed at room temperature for 1 hr, washed,

embedded, and cryosectioned as described (*Hutchins and Bronner, 2018*; *Hutchins and Bronner, 2019*). Sections were processed using a DuoLink PLA kit (Millipore/Sigma) with Anti-Rabbit MINUS PLA probe, anti-Goat PLUS PLA probe, and Far Red PLA detection reagent, and primary antibodies for Elavl1 and GFP, according to the manufacturer's DuoLink PLA Fluorescence protocol.

## Image acquisition and analysis

Confocal images were acquired using an upright Zeiss LSM 880 at the Caltech Biological Imaging Facility, and epifluorescence images were acquired using a Zeiss Imager.M2 with an ApoTome.2 module. Images were minimally processed for brightness/contrast and pseudocolored using Fiji (ImageJ, NIH) and Adobe Photoshop.

Relative fluorescence intensity was determined in Fiji. For each whole mount image, the line tool was used to draw an ROI surrounding the area of neural crest indicated by positive HCR fluorescence for the genes examined. For cross-sections, ROIs were drawn surrounding the neural crest and neural tube based on tissue morphology from nuclear staining. Following background subtraction (50-pixel rolling ball radius), integrated density was quantified for the ROIs on the control electroporated (left) and experimental electroporated (right) sides from the same embryo. Relative fluorescence intensity was then calculated by dividing the integrated density measurements for the experimental versus the control side of the same embryo.

Pax7 cell counts were performed as described (*Hutchins and Bronner, 2018*). The limit of the dorsal neural tube and characterization of cells as 'delaminated' or 'neural tube' was determined based on tissue morphology from nuclear staining. For relative migration distance determined from Pax7-stained embryos, distance of migration was measured in Fiji as described (*Hutchins and Bronner, 2018*).

For PLA analysis, confocal micrographs were processed in Fiji. Following background subtraction (50-pixel rolling ball radius), ROIs were drawn surrounding the neural crest and neural tube based on tissue morphology from nuclear staining, then puncta (PLA signal) were counted using 'analyze particles' function following global manual thresholding to reduce the appearance of nonspecific puncta outside of electroporated regions.

## Cloning of the *Gallus Axud1* 3′-UTR

Given that the 3′-UTR sequence of *Gallus Axud1* was not available in a public repository, to determine whether the *Axud1* 3′-UTR contained potential Elavl1-binding sites, we performed 3′-Rapid Amplification of cDNA Ends (3′-RACE) as described (*Scotto-Lavino et al., 2006*). Briefly, reverse transcription was performed using the $Q_T$ primer on total RNA extracted from HH8-9 embryos. First round PCR was performed on $Q_T$-primed cDNA using the $Q_O$ primer and *Axud1* Gene-Specific Primer 1. Second round PCR was then performed on this amplification product using the $Q_I$ primer and *Axud1* Gene-Specific Primer 2. The resulting PCR product was cloned into the pGEM-T Easy vector (Promega) and eight of these clones were sequenced, and the consensus 3′-UTR sequence deposited to GenBank (Accession # ON920861). Primer sequences can be found in the Key Resources Table.

## RNA structure and Elavl1-binding site prediction

Secondary structures for the *Draxin* 3′-UTR (GenBank: AB427147.1), *Axud1* 3′-UTR (GenBank: ON920861), *Msx1* 3′-UTR (Ensembl: ENSGALT00000024209.4), *BMP4* 3′-UTR (Ensembl: ENSGALT00000020316.7), *Pax7* 3′-UTR (Ensembl: ENSGALT00000048594.2), and *Tfap2b* 3′-UTR (Ensembl: ENSGALT00000026916.6) were predicted using 'mfold' web server with default settings (*Zuker, 2003*). Elavl1-binding sites were predicted using RBPDB (*Cook et al., 2011*) with 0.9 threshold.

## Statistical analyses

Statistical analyses were performed using Prism (8, 9; GraphPad Software). p values are defined in the text, and significance was established with $p < 0.05$. p values were calculated using Wilcoxon signed-rank, Mann–Whitney, one-sample *t*-tests, one-way ANOVA with post hoc Tukey, unpaired or paired *t*-tests as indicated; tests were two-tailed unless otherwise specified in the text/legend. Data were confirmed to be normally distributed using Kolmogorov–Smirnov or Kruskal–Wallis tests for parametric tests. Data measuring fluorescence intensities or cell counts for Experimental/Control sides are presented as box plots with individual data points shown. Bar graphs representing qPCR fold

enrichment are presented as mean values, with error bars indicating standard error of the mean. The number of embryos and replicates is indicated in figure legends and/or text. Post hoc power analyses (*Faul et al., 2007*) confirmed sufficient statistical power was reached (≥0.8) for reported p values and sample sizes.

## Acknowledgements

We thank A Collazo and G Spigolon for imaging assistance at the Caltech Biological Imaging Facility; M Schwarzkopf and G Shin (Molecular Technologies) for HCR probe design; I Antoshechkin of the Caltech Millard and Muriel Jacobs Genetics and Genomics Laboratory for sequencing of RNA-seq libraries; S Manohar for assistance with *Axud1* 3′-RACE; and G da Silva Pescador and R Galton for assistance with pilot experiments.

## Additional information

### Competing interests

Marianne E Bronner: Senior editor, *eLife*. The other authors declare that no competing interests exist.

### Funding

| Funder | Grant reference number | Author |
| --- | --- | --- |
| National Institute of Dental and Craniofacial Research | R01DE027538 | Marianne E Bronner |
| Amgen Foundation | Caltech Amgen Scholars Program | Jose Chacon |
| California State University, Northridge | BUILD PODER Program TL4GM118977 | Jose Chacon |
| National Institute of Dental and Craniofacial Research | K99DE029240 | Michael Piacentino |
| National Institute of Dental and Craniofacial Research | K99DE028592 | Erica J Hutchins |
| National Institute of Dental and Craniofacial Research | R01DE027568 | Marianne E Bronner |

The funders had no role in study design, data collection, and interpretation, or the decision to submit the work for publication.

### Author contributions

Erica J Hutchins, Conceptualization, Formal analysis, Investigation, Visualization, Methodology, Writing – original draft; Shashank Gandhi, Formal analysis, Visualization, Writing – review and editing; Jose Chacon, Investigation, Visualization, Writing – review and editing; Michael Piacentino, Formal analysis, Methodology, Writing – review and editing; Marianne E Bronner, Conceptualization, Supervision, Funding acquisition, Writing – review and editing

### Author ORCIDs

Erica J Hutchins http://orcid.org/0000-0002-4316-0333
Shashank Gandhi http://orcid.org/0000-0002-4081-4338
Jose Chacon http://orcid.org/0000-0001-7965-3976
Michael Piacentino http://orcid.org/0000-0003-1773-031X
Marianne E Bronner http://orcid.org/0000-0003-4274-1862

### Decision letter and Author response

Decision letter https://doi.org/10.7554/eLife.63600.sa1
Author response https://doi.org/10.7554/eLife.63600.sa2

## Additional files

### Supplementary files

• Transparent reporting form

### Data availability

RNA-sequencing datasets have been deposited on NCBI under the accession number PRJNA861325. The 3' untranslated region (UTR) sequence for Axud1 has been deposited to GenBank under accession number ON920861.

The following datasets were generated:

| Author(s) | Year | Dataset title | Dataset URL | Database and Identifier |
|---|---|---|---|---|
| Hutchins EJ, Gandhi S, Chacon J, Piacentino ML, Bronner ME | 2022 | RNA-binding protein Elavl1/HuR is required for maintenance of cranial neural crest specification | https://www.ncbi.nlm.nih.gov/nuccore/ON920861 | NCBI GenBank, ON920861 |
| Hutchins EJ, Gandhi S, Chacon J, Piacentino ML, Bronner ME | 2022 | RNA-binding protein Elavl1/HuR is required for maintenance of cranial neural crest specification | https://www.ncbi.nlm.nih.gov/bioproject/?term=PRJNA861325 | NCBI BioProject, PRJNA861325 |

The following previously published dataset was used:

| Author(s) | Year | Dataset title | Dataset URL | Database and Identifier |
|---|---|---|---|---|
| Williams RM, Candido-Ferreira I, Repapi E, Gavriouchkina D, Senanayake U, Telenius J, Ling I, Taylor S, Hughes J, Sauka-Spengler T | 2019 | Reconstruction of the global neural crest gene regulatory network in vivo | https://www.ncbi.nlm.nih.gov/geo/query/acc.cgi?acc=GSE121527 | NCBI Gene Expression Omnibus, GSE121527 |

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

# Appendix 1

## Appendix 1—key resources table

| Reagent type (species) or resource | Designation | Source or reference | Identifiers | Additional information |
|---|---|---|---|---|
| Gene (*Gallus gallus*) | *Elavl1* | GenBank | NM_204833.1 | |
| Gene (*Gallus gallus*) | *Draxin* | GenBank | AB427147.1 | |
| Gene (*Gallus gallus*) | *Axud1 (CSRNP1)* | This paper | GenBank: ON920861 | 3'-Untranslated region |
| Gene (*Gallus gallus*) | *Msx1* | Ensembl | ENSGALT00000024209.4 | |
| Gene (*Gallus gallus*) | *BMP4* | Ensembl | ENSGALT00000020316.7 | |
| Gene (*Gallus gallus*) | *Tfap2b* | Ensembl | ENSGALT00000026916.6 | |
| Gene (*Gallus gallus*) | *Pax7* | Ensembl | ENSGALT00000048594.2 | |
| Strain, strain background (*Gallus gallus*) | Rhode Island Red | Sun State Ranch (Monrovia, CA, USA) | | |
| Antibody | Rabbit anti-Elavl1 | Abcam | Cat# ab196626 | 1:100 immunohistochemistry; 5 µg immunoprecipitation |
| Antibody | Mouse IgG1 anti-Pax7 | Developmental Studies Hybridoma Bank | Cat# pax7, RRID:AB_528428 | 1:5 |
| Antibody | Goat IgG anti-GFP | Rockland | Cat# 600-101-215, RRID:AB_218182 | 1:500 |
| Antibody | Rabbit IgG, polyclonal – Isotype Control (ChIP Grade) | Abcam | Cat# ab171870 | 5 µg immunoprecipitation |
| Recombinant DNA reagent | pCI-H2B-RFP (plasmid) | *Betancur et al., 2010* | N/A | |
| Recombinant DNA reagent | pCIG (plasmid) | *Megason and McMahon, 2002* | N/A | |
| Recombinant DNA reagent | Draxin-FLAG (plasmid) | *Hutchins and Bronner, 2018* | N/A | |
| Recombinant DNA reagent | MCP-GFP (plasmid) | *Hutchins et al., 2020* | N/A | |
| Recombinant DNA reagent | MS2-Draxin 3'-UTR (plasmid) | This paper; *Hutchins et al., 2020* | N/A | |
| Sequence-based reagent | Control morpholino | GeneTools | N/A | 5'-CCTCTTACCTCAGTTACAAT TTATA |
| Sequence-based reagent | Elavl1 morpholino | This paper; GeneTools | N/A | 5'-GACATCTTATAACGTATCTC GCTGC |
| Sequence-based reagent | Draxin morpholino | *Hutchins and Bronner, 2018*; GeneTools | N/A | 5'-AAGGTGGAAGAAGCTGCCAT AATCC |
| Sequence-based reagent | Draxin qPCR primers | This paper; IDT | Custom DNA oligos | Forward: 5'-CTACGCTGTTATGCCA AATTCC; Reverse: 5'-GAATGATCCCTGCTCT CCATT |
| Sequence-based reagent | Axud1 qPCR primers | This paper; IDT | Custom DNA oligos | Forward: 5'-TCCAGTCCTTCTCGGA CTATAA; Reverse: 5'-GGGAAATTAGGCAACT GAAACTG |
| Sequence-based reagent | Msx1 qPCR primers | This paper; IDT | Custom DNA oligos | Forward: 5'-AGCTGGAGAAGC TGAAGATG; Reverse: 5'-AGGC TCCGTACAGGGAT |
| Sequence-based reagent | BMP4 qPCR primers | This paper; IDT | Custom DNA oligos | Forward: 5'-TTGGTCAACTCCGTCA ACTC; Reverse: 5'-CAGCACCACCTTGTCA TACT |
| Sequence-based reagent | Pax7 qPCR primers | This paper; IDT | Custom DNA oligos | Forward: 5'-CAAACCAACTCG CAGCATTC; Reverse: 5'-CTGCCTCCATCTTGGG AAAT |

*Appendix 1 Continued on next page*

*Appendix 1 Continued*

| Reagent type (species) or resource | Designation | Source or reference | Identifiers | Additional information |
|---|---|---|---|---|
| Sequence-based reagent | FoxD3 qPCR primers | This paper; IDT | Custom DNA oligos | Forward: 5'-CATCTGCGAGTTCATC AGCA; Reverse: 5'-TTCACGAAGCAGTCGT TGAG |
| Sequence-based reagent | 3'-RACE QT primer | *Scotto-Lavino et al., 2006*; IDT | Custom DNA oligos | 5'-CCAGTGAGCAGAGTGACGAG GACT CGAGCTCAAGCTTTTTTTTTTTTT TTTT |
| Sequence-based reagent | 3'-RACE QO primer | *Scotto-Lavino et al., 2006*; IDT | Custom DNA oligos | 5'-CCAGTGAGCAGAGTGACG |
| Sequence-based reagent | 3'-RACE QI primer | *Scotto-Lavino et al., 2006*; IDT | Custom DNA oligos | 5'-GAGGACTCGAGCTCAAGC |
| Sequence-based reagent | 3'-RACE Axud1 Gene Specific Primer 1 | This paper; IDT | Custom DNA oligos | 5'-CGTGTTCCAAGAGCTATGCC |
| Sequence-based reagent | 3'-RACE Axud1 Gene Specific Primer 2 | This paper; IDT | Custom DNA oligos | 5'-GGTTTCCCGCAAGCTGG |
| Commercial assay or kit | RNAqueous-Micro Total RNA isolation kit | Ambion | Cat# AM1931 | |
| Commercial assay or kit | SMART-Seq v4 Ultra Low Input cDNA kit | Takara Bio | Cat# 634889 | |
| Commercial assay or kit | Agencourt AMPure XP beads | Beckman Coulter | Cat# A63880 | |
| Commercial assay or kit | Endofree maxi prep kit | Qiagen | Cat# 12362 | |
| Commercial assay or kit | Duolink In Situ PLA Probe Anti-Rabbit MINUS | Millipore/Sigma | Cat# DUO92005 | |
| Commercial assay or kit | Duolink In Situ PLA Probe Anti-Goat PLUS | Millipore/Sigma | Cat# DUO92003 | |
| Commercial assay or kit | Duolink In Situ Detection Reagents FarRed | Millipore/Sigma | Cat# DUO92013 | |
| Commercial assay or kit | Duolink In Situ Wash Buffers, Fluorescence | Millipore/Sigma | Cat# DUO82049 | |
| Chemical compound, drug | Duolink In Situ Mounting Medium with DAPI | Millipore/Sigma | Cat# DUO82040 | |
| Chemical compound, drug | Fluoromount-G | SouthernBiotech | Cat# 0100-01 | |
| Chemical compound, drug | DAPI | Thermo Fisher | Cat# D1306 | 1:5000 |
| Chemical compound, drug | FastStart Universal SYBR Green Master (Rox) | Millipore/Sigma | Cat# FSUSGMMRO | |
| Chemical compound, drug | SuperScript III Reverse Transcriptase | Thermo Fisher | Cat# 18080044 | |
| Chemical compound, drug | cOmplete, Mini, EDTA-free Protease Inhibitor | Millipore/Sigma | Cat# 11836170001 | |
| Chemical compound, drug | RNaseOUT Recombinant Ribonuclease Inhibitor | Thermo Fisher | Cat# 10777019 | |
| Chemical compound, drug | Accumax | Innovative Cell Technologies, Inc | Cat# AM105 | |
| Chemical compound, drug | Protein G Dynabeads | Thermo Fisher | Cat# 10003D | |
| Chemical compound, drug | Actinomycin D | Millipore/Sigma | Cat# A9415 | 10 µg/ml |
| Software, algorithm | Fiji | *Schindelin et al., 2012* | RRID:SCR_002285 | https://imagej.net/Fiji |
| Software, algorithm | Seurat | *Butler et al., 2018* | RRID:SCR_007322 | https://satijalab.org/seurat/ |

*Appendix 1 Continued on next page*

*Appendix 1 Continued*

| Reagent type (species) or resource | Designation | Source or reference | Identifiers | Additional information |
|---|---|---|---|---|
| Software, algorithm | QuantStudio Design & Analysis software, version 2.4 | Life Technologies | RRID:SCR_018712 | |
| Software, algorithm | Zen 2 Blue | Zeiss | N/A | |
| Software, algorithm | Zen Black | Zeiss | N/A | |
| Software, algorithm | Photoshop CC | Adobe | N/A | |
| Software, algorithm | Prism8, Prism9 | GraphPad | N/A | |
| Software, algorithm | FastQC | *Andrews, 2014* | RRID:SCR_014583 | |
| Software, algorithm | Cutadapt | *Martin, 2011* | RRID:SCR_011841 | |
| Software, algorithm | Bowtie2 | *Langmead and Salzberg, 2012* | RRID:SCR_016368 | |
| Software, algorithm | HTSeq-Count | *Anders et al., 2015* | RRID:SCR_011867 | |
| Software, algorithm | biomaRt package | *Durinck et al., 2009* | RRID:SCR_019214 | |
| Software, algorithm | mfold | *Zuker, 2003* | RRID:SCR_008543 | |
| Software, algorithm | G*Power | *Faul et al., 2007* | RRID:SCR_013726 | |
| Other, deposited data | sc-RNA (SmartSeq2) | *Williams et al., 2019* | GEO: GSE130500 | Single-cell RNA-sequencing datasets; see *Figure 1* and Data Availability Statement |
| Other, deposited data | RNA-seq data (bulk) | This paper | NCBI Bioproject: PRJNA861325 | Single-end RNA-sequencing datasets; see *Figure 4* and Data Availability Statement |

