## [Editor Report]

In this short report, Hutchins et al. reveal expression of the RNA-binding protein (RBP) HuR in the neural tube and cranial neural crest of chicken embryos. Knock-down of HuR affects expression of Axud1 and FoxD3 (both genes associated with neural crest specification) and of the Wnt antagonist Draxin previously shown by the authors to regulate neural crest specification and delamination. The authors propose that HuR associates with Draxin mRNA and demonstrate that Draxin overexpression can rescue FoxD3 expression upon HuR knock down. The data is in line with the idea that control of neural crest specification by HuR at least partially involves Draxin mRNA stabilization.

---

## [Decision Letter]

**Decision letter after peer review:**

Thank you for submitting your article "RNA-binding protein Elavl1/HuR is required for maintenance of cranial neural crest specification" for consideration by *eLife*. Your article has been reviewed by 2 peer reviewers, and the evaluation has been overseen by a Reviewing Editor and Kathryn Cheah as the Senior Editor. The following individual involved in review of your submission has agreed to reveal their identity: Lukas Sommer (Reviewer #1).

The reviewers have discussed the reviews with one another and the Reviewing Editor has drafted this decision to help you prepare a revised submission.

Summary:

In this short report, Hutchins et al., reveal expression of the RNA-binding protein (RBP) HuR in the neural tube and cranial neural crest of chicken embryos. Knock-down of HuR affects expression of Axud1 and FoxD3 (both genes associated with neural crest specification) and of the Wnt antagonist Draxin previously shown by the authors to regulate neural crest specification and delamination. The authors propose that HuR associates with Draxin mRNA and demonstrate that Draxin overexpression can rescue FoxD3 expression upon HuR knock down. The data is in line with the idea that control of neural crest specification by HuR at least partially involves Draxin mRNA stabilization.

Essential revisions:

As outlined below in the largely unedited Reviewers comments, they suggest a major re-write of the paper.

Overall, Reviewer 1 commented that unraveling the gene regulatory network (GRN) of neural crest specification has been the focus of many recent studies. How the GRN is modulated by post-transcriptional control mechanisms, as addressed here, is an important and relevant question. However they feel that the authors missed an opportunity to address this question in a broader manner. Moreover, the evidence that HuR regulates Draxin post-transcriptionally (by mRNA stabilization) has to be strengthened.

Overall, Reviewer 2 commented that the main problem about this work is that the central player of the story seems to be a ubiquitous expressed molecule (a paper in the literature mentions Elavl1 as "universally" expressed). In the early chicken embryo, the situation does not seem very different, since it is certainly expressed in forming NC/NT, but also in other tissues (ectoderm, mesoderm, Figure 1). Thus reader might conclude that it is not very surprising that when mRNAs are less stable, most tissues, including NC, do not perform as expected. Indeed they have shown that the three genes they have tested (Axud1, FoxD3 and Draxin) are all down-regulated after Elavl1 knock-down.

Major Points:

1) According to the Abstract and in the Introduction, the authors' goal of this study was the identification of potential post-transcriptional regulators of neural crest specification. Specifically, they "sought to identify RBPs with early roles in neural crest development". This is an intriguing research question. However, rather than systematically assessing RBPs in their RNA seq datasets, they picked Elav1/HuR for their further experiments. Why? Likely, this was not the only RBP in their RNA seq datasets. Was Elav1 (encoding HuR) the most enriched RBP? These data should be shown and the reason for picking Elav1/HuR should be clarified.

2) In Figure 1D', expression of HuR in migratory neural crest is not evident. Transverse section at HH9+ might help revealing HuR expression in migratory neural crest cells.

3) The link between HuR and Draxin is rather weak and seems again a bit 'cherry picked'. I recommend a more global assay to identify mRNA targets of HuR (e.g. CLIP seq). As correctly stated in the Discussion, HuR is also known to stabilize Snail1 and other mRNAs with roles in neural crest development, which could all mediate post-transcriptional regulation of neural crest specification and, therefore, mediate the observed phenotypes. Importantly, in Figure 3, enrichment of Draxin mRNA is not very convincing, in particular when compared to its control (non-specific control IP) rather than to Pax7/FoxD3 values (which for reasons unclear to me are highly negative). A positive control should be included (e.g. a previously identified mRNA target of HuR). Can Draxin mRNA binding by HuR be confirmed by other means (such as CLIP seq)? It should be addressed whether Draxin mRNA stability is indeed affected by HuR knock down.

4) The Draxin overexpression rescue experiment is nice, but should be extended to other parameters of neural crest specification (the authors only checked FoxD3 expression). What about other markers? What about delamination?

5) The authors show that the three genes they have tested (Axud1, FoxD3 and Draxin) are all down-regulated after Elavl1 knock-down. The experiment shown in Figure 4 somewhat dampens this impression by suggesting that not all those mRNAs bind to Elavl1, but it would give weight to these findings if the authors would i) support this data by checking in silico on the many RNAseq data they have gathered over the years that only a subpopulation of "NC genes" display Elavl1 target sequences in their UTR and ii) verify that some of those genes are unaffected after Elavl1 MO electroporation (using the same approach than in Figure 2). This would support the idea that even though Elavl1 may be ubiquitously expressed, its targets are restricted and that we are looking at a regulated biological process and not so much at a house-keeping activity.

If these authors can show some specificity in Elavl1 function, this could also be an opportunity to re-write part of the text and convey a more general idea/concept on the importance of RNA stability in biology, something that we use on a daily basis when designing expression vectors, but which has obviously real-life implications.

6) It would also help if the authors use Elavl1 gain-of-function approaches to support the data presented here (would it over-stabilize Draxin, for instance?).

7) Very little (in fact nothing) is said in the introduction about existing work on full and conditional KO of Elavl1 in mouse (Katsanou et al., 2009), which sends the message that the present work stands on its own. Katsanou's paper is in fact quite extensive and it describes a number of phenotypes that the authors observed in different genetic settings. This has to be included in the present paper (i.e. it should be introduced/discussed upfront), with the aim of convincing the reader that there is still something to be understood about the function of Elavl1 and that the (advantages of the model) chicken embryo can address those questions in a more precise way than the murine model does.

---

## [Author Response]

Essential Revisions:1) According to the Abstract and in the Introduction, the authors' goal of this study was the identification of potential post-transcriptional regulators of neural crest specification. Specifically, they "sought to identify RBPs with early roles in neural crest development". This is an intriguing research question. However, rather than systematically assessing RBPs in their RNA seq datasets, they picked Elav1/HuR for their further experiments. Why? Likely, this was not the only RBP in their RNA seq datasets. Was Elav1 (encoding HuR) the most enriched RBP? These data should be shown and the reason for picking Elav1/HuR should be clarified.

We acknowledge the reviewers’ point, which made us realize that we did not adequately explain why we chose to focus on Elav1/HuR as a post-transcriptional regulator of neural crest development. We now clarify that this is one of the few such genes that we have found enriched in the premigratory neural crest. To better demonstrate this, we have re-analyzed a recently published scRNA-seq dataset (Williams et al., 2019). We have reorganized the paper and now include this analysis as a new Figure 1 to further clarify our choice for selecting Elavl1 for follow up.

In the revised manuscript, we highlight that several RNA-binding proteins are expressed to varying degrees in cranial neural crest cells. Using gene ontology analysis, we generated a three-way Venn diagram to identify overlap between genes associated with the gene ontology terms “binds RNA,” “regulates translation,” and “stabilizes RNA.” Only three genes, Elavl1, Dazl, and Igf2bp1, were associated with all three, and only Elavl1 is abundant amongst all sorted cranial neural crest cells.

We have now included this expanded justification in the text.

2) In Figure 1D', expression of HuR in migratory neural crest is not evident. Transverse section at HH9+ might help revealing HuR expression in migratory neural crest cells.

We have added a transverse section at HH9+ showing Elavl1 expression in migratory neural crest cells. This is now included in Figure 2F-G.

3) The link between HuR and Draxin is rather weak and seems again a bit 'cherry picked'. I recommend a more global assay to identify mRNA targets of HuR (e.g. CLIP seq).

We thank the reviewers for raising this good point. To more broadly identify targets of

Elavl1, we performed differential RNA-sequencing analysis on dissected neural folds (representing premigratory neural crest) for control vs Elavl1 knockdown (Figure 4). We identified 24 differentially expressed genes (12 downregulated, 12 upregulated). Among those genes, only four were known to have roles in neural crest development and were found to be significantly downregulated with Elavl1 knockdown—*Draxin*, *Axud1*, *Msx1*, and *BMP4*. The downregulation of these transcripts with Elavl1 knockdown was confirmed in vivo using HCR (Figure 3, Figure 4-Supplement 1).

As correctly stated in the Discussion, HuR is also known to stabilize Snail1 and other mRNAs with roles in neural crest development, which could all mediate post-transcriptional regulation of neural crest specification and, therefore, mediate the observed phenotypes.

The reviewers raise an important point. However, Snail1 is not expressed in chick cranial neural crest at these stages (del Barrio and Nieto, 2002, Development, 129(7):1583-93); rather Snail2 is the predominant Snail family transcription factor at this time and place in chick embryos. Importantly, our newly added RNA-seq data and HCR for Snai2 (Figure 4-Supplement 1) indicate that, indeed, Snai2 expression is unaffected by Elavl1 knockdown, and is unlikely to be a direct target of Elavl1 at the premigratory stage.

Importantly, in Figure 3, enrichment of Draxin mRNA is not very convincing, in particular when compared to its control (non-specific control IP) rather than to Pax7/FoxD3 values (which for reasons unclear to me are highly negative). A positive control should be included (e.g. a previously identified mRNA target of HuR). Can Draxin mRNA binding by HuR be confirmed by other means (such as CLIP seq)?

Thank you for this comment, which made us realize that this experiment needed to be better documented. Accordingly, we have modified the presentation of this RIP data to display Fold Enrichment, in keeping with other studies that have performed RIP experiments for Elavl1 (Rothamel et al., 2021). We hope this better illustrates the enrichment of *Draxin* mRNA over non-targets *FoxD3* and *Pax7*.

While it would be ideal to use an experiment such as CLIP-seq to confirm direct binding, CLIP-seq (and similar cross-linking experiments) require large amounts of starting material that make this approach unfeasible for embryonic tissue. However, in an effort to provide demonstration of *Draxin* mRNA binding by Elavl1 through other means, we have now included a proximity ligation assay experiment (Figure 4G) as a secondary means to demonstrate a direct interaction. Proximity ligation is a well-established technique used to demonstrate direct protein-protein interactions. We have now adapted this method to interrogate a protein-RNA interaction. For this experiment, we created an expression construct containing the Luciferase coding sequence followed by MS2 stem loops and the *Draxin*-3’UTR. When transcribed, and co-expressed with an MCP-GFP construct, the MS2 stem loops in the mRNA transcript are bound by MCP-GFP. Thus, if Elavl1 binds to the *Draxin*-3’UTR, a proximity ligation assay using antibodies to Elavl1 and GFP will generate fluorescent puncta. Using this approach, we now demonstrate that endogenous Elavl1 physically interacts with the 3’UTR of *Draxin* (Figure 4H-I).

It should be addressed whether Draxin mRNA stability is indeed affected by HuR knock down.

Good point. To quantify changes in mRNA stability with Elavl1 knockdown, we utilized the transcriptional inhibitor actinomycin D followed by qRT-PCR to measure RNA decay for *Draxin*, *Axud1*, *Msx1*, and *BMP4*, since we found using RNA-sequencing that these four genes are all downregulated with Elavl1 knockdown. We also examined RNA decay for *Pax7* as a “non-target”. Interestingly, we found that only *Draxin* mRNA stability was significantly decreased with Elavl1 knockdown (Figure 4C-D).

4) The Draxin overexpression rescue experiment is nice, but should be extended to other parameters of neural crest specification (the authors only checked FoxD3 expression). What about other markers? What about delamination?

We extended the rescue experiment to also examine delamination (as we had for Figure 3 with Elavl1 knockdown). We found that Draxin overexpression was sufficient to rescue defects in delamination (Figure 5K-L). Further, we also performed HCR for *Axud1*, *Msx1*, and *BMP4* (in addition to *FoxD3*), since we found these transcripts to also be downregulated with Elavl1 knockdown. All of these transcripts were rescued from Elavl1 knockdown with Draxin overexpression (Figure 5F-J). Importantly, Draxin knockdown alone also significantly reduced expression of these same four transcripts (Figure 5A-E). Together these data support the hypothesis that *Draxin* is the major target of Elavl1 in cranial neural crest specification. These data are included in Figure 5.

5) The authors show that the three genes they have tested (Axud1, FoxD3 and Draxin) are all down-regulated after Elavl1 knock-down. The experiment shown in Figure 4 somewhat dampens this impression by suggesting that not all those mRNAs bind to Elavl1, but it would give weight to these findings if the authors would i) support this data by checking in silico on the many RNAseq data they have gathered over the years that only a subpopulation of "NC genes" display Elavl1 target sequences in their UTR and ii) verify that some of those genes are unaffected after Elavl1 MO electroporation (using the same approach than in Figure 2). This would support the idea that even though Elavl1 may be ubiquitously expressed, its targets are restricted and that we are looking at a regulated biological process and not so much at a house-keeping activity.If these authors can show some specificity in Elavl1 function, this could also be an opportunity to re-write part of the text and convey a more general idea/concept on the importance of RNA stability in biology, something that we use on a daily basis when designing expression vectors, but which has obviously real-life implications.

The reviewers raise excellent points. We now include Elavl1 binding analysis for the 3’UTRs of *Draxin*, *Axud1*, *Msx1*, *BMP4*, *Pax7*, and *Tfap2b* (Figure 4-Supplement 2). Neither *Axud1*, *Pax7* nor *Tfap2b* contain Elavl1 binding sites, and *Msx1* and *BMP4* contain only one. Notably, the *Draxin*-3’UTR contains four high-probability binding sites, all in proximity on the same side of the folded structure.

Given that Elavl1 contains three RNA-recognition motifs which are known to cooperatively bind a target RNA (Pabis et al., 2018), it is possible that, in order for Elavl1 to bind and stabilize an mRNA target, it needs to establish multiple points of contact on the RNA, which may account for the specificity of function we have observed here in neural crest.

In addition to the new data figure, we also expand on the above points in the relevant section of the Results.

6) It would also help if the authors use Elavl1 gain-of-function approaches to support the data presented here (would it over-stabilize Draxin, for instance?).

We overexpressed Elavl1 in premigratory cranial neural crest cells (driven by the FoxD3 enhancer NC1) and performed HCR for *Draxin*; however, we did not observe differences in *Draxin* levels between the control and Elavl1 overexpressing sides at HH9. We chose not to include these data in the manuscript as we do not think it adds anything to or changes the interpretation of our data or conclusions. This finding is consistent with previous observations in HeLa cells, where reduction but not overexpression of Elavl1 altered type I IFN response (Herdy et al., 2015, Eur J Immunol 45(5):1500-11. doi: 10.1002/eji.201444979).

7) Very little (in fact nothing) is said in the introduction about existing work on full and conditional KO of Elavl1 in mouse (Katsanou et al., 2009), which sends the message that the present work stands on its own. Katsanou's paper is in fact quite extensive and it describes a number of phenotypes that the authors observed in different genetic settings. This has to be included in the present paper (i.e. it should be introduced/discussed upfront), with the aim of convincing the reader that there is still something to be understood about the function of Elavl1 and that the (advantages of the model) chicken embryo can address those questions in a more precise way than the murine model does.

Apologies that we did not sufficiently reference this paper. While our original draft referenced this paper in the Introduction and Results sections with respect to their findings in craniofacial development, which we felt was most pertinent, in the revised manuscript we have now expanded reference to this study in the Introduction. In light of our newly added results, we have also now added reference to this citation in the Discussion, where we describe our findings, how they relate to and build upon the Katsanou paper, and how the chick model enables us to ask these questions in a spatiotemporally controlled manner.